# Growth Trend Prediction and Intervention of Panax Notoginseng Growth Status Based on a Data-Driven Approach

**DOI:** 10.3390/plants14081226

**Published:** 2025-04-16

**Authors:** Jiahui Ye, Xiufeng Zhang, Gengen Li, Chunxi Yang, Qiliang Yang, Yuzhe Shi

**Affiliations:** 1Faculty of Mechanical and Electrical Engineering, Kunming University of Science and Technology, Kunming 650500, China; yjh123@stu.kust.edu.cn (J.Y.); 18810575316@163.com (X.Z.); lg15571195216@163.com (G.L.); 2Yunnan Key Laboratory of Intelligent Control and Application, Kunming University of Science and Technology, Kunming 650500, China; 3Faculty of Modern Agricultural Engineering, Kunming University of Science and Technology, Kunming 650500, China; shiyuzhe9@163.com

**Keywords:** Panax notoginseng, plant growth, irrigated cultivation, smart agriculture

## Abstract

In crop growth, irrigation has to be adjusted according to developmental stages. Smart agriculture requires the accurate prediction of growth status and timely intervention to improve the quality of agricultural products, but this task faces significant challenges due to variable environmental factors. To address this issue, this study proposes a data-driven irrigation method to enhance crop yield. Our approach harvests extensive datasets to train and optimize an integrated deep-learning architecture combining Informer, Long Short-Term Memory (LSTM) networks, and Exponential Weighted Moving Average (EWMA) models. Controlled greenhouse experiments validated the reliability and practicality of the proposed prediction and intervention strategy. The results showed that the model accurately issued irrigation warnings 3–5 days in advance. Compared to traditional fixed irrigation, the model significantly reduced irrigation frequency while maintaining the same or even better growth conditions. In terms of plant quantity, the experimental group increased by 410.0%, while the control group grew by 50.0%. Additionally, the experimental group’s average plant height was 21.8% higher than that of the control group. These results demonstrate the efficacy of the proposed irrigation prediction method in enhancing crop growth and yield, providing a novel strategy for future agricultural planning and management.

## 1. Introduction

To advance modern agriculture, China has implemented a series of policies aimed at supporting agricultural modernization [1,2]. These irrigation modernization policies play a crucial role in promoting sustainable strategies for developing countries [3]. In recent years, many scholars have adopted deep-learning- or data-driven-based approaches to study various types of crops in order to improve the yield of crops. One of the keys to increasing crop yields is controlling the amount of water irrigated at the planting field [4].

Moisture has an important effect on crop growth, and different amounts of irrigation are required at different growth stages to maintain crop health [5]. In addition, different irrigation methods can have different effects on crops [6]. In order to obtain the characteristics of soil irrigation systems, mobile platforms and data processing algorithms can be developed in conjunction with portable environmental sensors [7]. What is more, the effective combination of deep learning and IoT can optimize the irrigation strategy of the planting site so as to increase the crop yield [8,9].

Many different crop-growth models have been developed based on different climatic conditions such as light and temperature [10,11]. These traditional approaches are knowledge based and estimate crop growth based on external factors [12]. However, in modern agriculture, obtaining the right intervention time at the planting field is still faced with great challenges. This is why some scholars have started to study the growth trend of crops in order to increase the yield of crops in the planted field [13]. In order to help the farmer find the right time to take action on their planting sites (irrigation, weeding, etc.), plant growth is predicted by neural networks, and this is used as a basis for farming strategies [14]. By predicting the shape of future images of plants through a plant-growth prediction algorithm based on deep learning and spatial transformation, predicting the growth behavior of plants can help managers perform a timely intervention [15]. But the process of plant growth and development is highly unstable in many aspects. For example, the spatial correlation or temporal dependence of localized pixel values in plant images are factors that seriously affect the accuracy of prediction [16].

The extensive use of machine learning in agriculture, especially deep-learning methods, can predict crop status early and provide appropriate intervention measures [17]. Predictive modeling that combines IoT with Long Short-Term Memory (LSTM) networks can effectively help farmers take timely protective measures for their crops [18]. Early prediction and sensing of crop status ensures safety for subsequent planning and enables farmers to plan with comprehensive spatial and temporal information [19]. The data decision scheme is conducive to the improvement of agricultural productivity and product quality [20]. In modern agriculture, there are many models of crop growth of different types and in various environments, which include a variety of environmental factors along with human management factors [21]. Since the growth of crops is affected by many external conditions, it is very challenging to predict the growth state of crops [15].

In the study of agricultural irrigation strategies, traditional inefficient irrigation methods have been proven to significantly constrain crop yields [22]. In recent years, the application of smart irrigation systems has provided new solutions for improving water-use efficiency [23]. Research has shown that irrigation strategies optimized based on differential evolution algorithms can significantly enhance crop yields [24]. Furthermore, by integrating the CERES-Wheat model with a non-dominated sorting genetic algorithm, not only has the optimization of irrigation strategies been achieved, but accurate prediction of crop yields has also been realized [25]. Moreover, scholars have introduced an innovative Spatial Fuzzy Strategic Planning (SFSP) framework, integrating Multi-Criteria Decision-Making (MCDM) with a conceptual agricultural water-use model to systematically support sustainable agricultural water management strategies [26].

However, these methods primarily optimize irrigation strategies based on soil moisture status, failing to adequately address the growth requirements of the crops themselves. To address this limitation, this study proposes a dynamic irrigation approach based on crop-growth status. By real-time monitoring of crop-growth conditions and adjusting irrigation schemes, this method achieves more precise and efficient irrigation management.

To validate this crop-centric strategy, we focus on Panax notoginseng, a high-value medicinal crop in China with documented pharmacological significance [27,28] that exhibits hypersensitive responses to soil moisture dynamics [29]. To maximize yield and sustain commercial viability, optimized cultivation practices are critical. Recent advancements in greenhouse cultivation technology [30] offer promising solutions for precise environmental control, particularly in managing critical growth parameters. This study therefore employs Panax notoginseng as a model crop to investigate how greenhouse-based interventions, specifically those integrating soil moisture regulation and controlled growing conditions, can enhance productivity while addressing the plant’s ecological constraints. The research aims to establish a framework for targeted agricultural management, ultimately supporting the sustainable development of this pharmaceutically valuable species.

In this paper, environmental data of the planting field and morphological data of the crop in a glass greenhouse are collected. Then, the data is combined with a proposed Informer–LSTM–EWMA approach to predict the growth trend of the crop. The trend prediction of Panax notoginseng enables the manager to be notified to take irrigation measures in advance. The proposed method has the following features:*Multi-source Data Fusion:* Real-time collection of environmental data such as soil, temperature, and humidity through sensors, combined with IoT technology for precise perception.*Lifecycle Prediction and Optimization:* Integration of historical and real-time data to predict the growth stages of Panax notoginseng using the Informer–LSTM–EWMA method, optimizing management strategies.*Irrigation optimization:* An event-triggered irrigation strategy for water conservation and efficiency enhancement.

Here, we used the plant height of the entire experimental field during the growing period as the phenotype. The past environmental data and morphological data of the crop are used as inputs to predicate the future growth trend of the crop. The main contributions of this paper are as follows:A distributed detection system for greenhouse environmental data acquisition collects various parameters and transmits them to the cloud via 4G for storage and display.The Informer–LSTM–EWMA model is developed to predict Panax notoginseng growth trends, leveraging historical data to accurately forecast intervention timings, ensuring sufficient time for action.This study optimizes intervention strategies based on an intervention timing prediction model and experimentally validates the significant effectiveness of this optimized approach in enhancing the yield of Panax notoginseng.

## 2. Materials

### 2.1. Description of the Experimental Field

Field experiments were conduct at the Tianyuan Panax notoginseng planting base of Kunming University of Science and Technology in Kunming City, Yunnan Province, China (latitude $24^{{}^{\circ}}50^{\prime }53^{\prime \prime }$ N, longitude $102^{{}^{\circ}}51^{\prime }48^{\prime \prime }$ E), from September 2023 to June 2024. The altitude of the experiment site is 2067 m. The location has a subtropical monsoon climate. The experimental site is shown in Figure 1.

The experimental site is characterize by a spring-like climate throughout the year, classified as a plateau mountain monsoon climate. The monthly average temperature ranges from 3 °C to 27 °C, with an average high of 22 °C and an average low of 12 °C. The annual average rainfall is 797.5 mm. The month with the highest relative humidity is July (83%), while the month with the lowest relative humidity is March (58%). The annual evaporation is between 1200 and 1500 mm. Each experimental field is a separate concrete recess, and the experimental fields did not interfere with one another. The transpiration rates of the crops are shown in Table 1. Because the experimental fields are set in a glass greenhouse (as seen in the right of Figure 1), the interference of the external environment to the shed can be ignored.

### 2.2. Data Acquisition System

In order to collect the environmental data of air and soil, an IoT-based data collection and transmission system is design in this experiment, as shown in Figure 2. The sensors collect data once per hour, resulting in 24 data points per day. The sensor nodes upload the collect data to local terminal devices via the Bluetooth communication protocol. Each terminal device is equipped with multi-channel access capability, enabling them to simultaneously receive data streams from multiple distribute sensor groups. After the data are transmitted to the cloud server via a 4G wireless network, authorized users can access the web-based visualization platform through identity authentication, achieving real-time monitoring of measurement parameters and historical data traceability.

During the experimental process, conditions such as soil and light are consistent across all experimental plots, with the timing of interventions and the amount of irrigation being the sole controlled variables. Soil and air sensors collected data hourly, while the morphological data of Panax notoginseng are measure weekly. Throughout the experiment, we accumulated approximately more than 6800 data entries. Each entry included soil temperature, moisture, and conductivity collected by various soil sensors; air temperature and humidity; temperature, moisture, and conductivity differences between different soil layers; and average temperature, moisture, and conductivity of adjacent sensors; as well as the morphological data of Panax notoginseng, such as plant height and stem diameter. A total of 4×5 soil sensors and one air sensor are arranged. The arrangement of the sensors is shown in Figure 3.

The sensor sends the collected data to the upper computer via a Bluetooth module. A computer can receive data from multiple sensors and uploads the data to the cloud via a 4G base station. Therefore, the user can access the cloud to view the data in real time. The workflow is shown in Figure 4.

The soil sensor collects soil temperature, soil moisture, and conductivity. The air sensor collects the temperature and humidity data of the air. All these parameters are collected by the system per hour. Sensor data acquisition parameters are shown in Table 2. The technical parameters of the air sensor and soil sensor are shown in Table 3 and Table 4, respectively.

Morphological measurements of Panax notoginseng plants are conducted weekly in each experimental field. Plant height is measure using a ruler, and stem diameter is determined at the stem bifurcation point using a vernier caliper. Each measurement is performed in triplicate, and the mean value is calculated as the final result [31]. All plants within the experimental fields are systematically measured. This experiment utilizes two-year-old Panax notoginseng as the research subject, with the experimental field adopting a double-row planting pattern. The row spacing is set at 0.3 m, and the plant spacing is controlled at 0.1 m. The average plant height per plot, crop survival rate, and growth rate are used as key evaluation metrics for comparative analysis.

### 2.3. Data Processing

Since the sensor in this experiment collects data, the collected data naturally contains some noise. In order to minimize the effect of noise, this paper uses Gaussian filtering [32] to reduce the noise of the data, as shown in Equation (Equation 1).(1)$$G\left(x\right)=\frac{1}{\sqrt{2\pi }\sigma }e^{-\frac{x^{2}}{2\sigma^{2}}}$$

Let G(x) represent the output signal after Gaussian denoising, *x* represent the value of the input signal, and σ represent the standard deviation of the Gaussian distribution. In this paper, σ is set to 10.

During the data collection process in the Panax notoginseng experimental field, anomalies may occur due to errors in data collection, storage, processing, noise during data capturing, or variations in data generation. These anomalous data points can negatively impact the accuracy and effectiveness of data analysis and machine-learning models. Therefore, this paper employs the Interquartile Range (IQR) method to detect and handle such data anomalies.

The Interquartile Range (IQR) method divides the data into quartiles. The first quartile, $Q_{1}$, also known as the “lower quartile”, corresponds to the value at the 25th percentile when all data points in the sample are arrange in ascending order. The second quartile, $Q_{2}$, also known as the “median”, corresponds to the value at the 50th percentile. The third quartile, $Q_{3}$, also known as the “upper quartile”, corresponds to the value at the 75th percentile. The difference between the third quartile and the first quartile is referred to as the Interquartile Range (IQR), also known as the quartile deviation. The calculation equation is shown in Equation (Equation 2).(2)$$IQR=Q_{3}-Q_{1}$$

Then, a threshold is set (in this paper, the threshold is set to 1.5IQR), and data points smaller than $Q_{1}-1.5IQR$ or larger than $Q_{3}-1.5IQR$ are considered outliers. The outlier determination mechanism follows a two-stage analytical workflow: statistical methods are initially applied to establish valid distribution boundaries (upper and lower limits), with subsequent observational data points exceeding these thresholds being formally classified as outliers. The data from 5 July 2023 to 13 July 2023 are select for outlier detection and analysis, as shown in Figure 5.

The red data points exceeding the defined boundaries in Figure 5 represent identified outliers. In our processing methodology, these anomalous values are systematically removed and replaced with the mean value of the dataset.

The handling of missing data in this study employed linear interpolation for imputation, with the mathematical formulation explicitly provided as show in Equation (Equation 3).(3)$$y_{k}=y_{k-1}+\left(t_{k}-t_{k-1}\right)\frac{y_{k+1}-y_{k-1}}{t_{k+1}-t_{k-1}}=y_{k-1}+\frac{\left(t_{k}-t_{k-1}\right)y_{k+1}-\left(t_{k}-t_{k-1}\right)y_{k-1}}{t_{k+1}-t_{k-1}}$$

In the equation, $y_{k}$ denotes the data value at time $t_{k}$. When the sensor data $y_{k}$ at time $t_{k}$ is missing, the missing value can be computed using Equation (Equation 3) by leveraging the known data $y_{k-1}$ from the preceding time instant $t_{k-1}$ and the known data $y_{k+1}$ from the subsequent time instant $t_{k+1}$.

### 2.4. Experimental Design and Management

In the experimental environment of the greenhouse, each Panax notoginseng field has a fixed amount of water irrigation every week. Traditional irrigation is carried out using a drip irrigation method, with watering conducted once a week, each time using approximately 135 L of water. The traditional fixed-water irrigation method does not give much consideration to the crop’s water demand. However, too much or too little irrigation has different effects on Panax notoginseng [33], and the crop has different water needs at different stages of the crop’s life. Unsuitable irrigation water can lead to slow growth or even serious crop death, influencing the product quality and yields of the planting field.

Within the experimental field, other external conditions, such as soil, light, etc., are kept at the same level in the experimental field, and the only control parameter is the amount of water.

In the experimental design stage, within-group control and between-group control are two common types of control, which have different characteristics and application scenarios. In addition to the mutual experimental control between the five test fields, this experiment divided each field into experimental and control groups (about 1/3 of the experimental group and 2/3 of the control group) to form a within-group control. The experimental field is divided as shown in Figure 6.

In Figure 6, we established five experimental fields numbered 1 to 5, each subjected to different irrigation strategies for treatment. The specific irrigation schemes for individual experiments are detailed in the subsequent Table 5 and its accompanying descriptions.

The experiment is designed with five experimental fields, and the timing of irrigation obtained by our proposed method in each field is shown in Table 5. The morphological data of total plant height and stem thickness of Panax notoginseng in each experimental field are selected as the status indicator. The accuracy of the proposed method and the rationality are verify by the growth trends before and after the intervention.

Within each experimental field, within-group controls are conducted. For example, A1 vs. A2, B1 vs. B2, C1 vs. C2, D1 vs. D2, and E1 vs. E2; the former used data-driven decision making to irrigate the experimental fields, while the latter used traditional fixed water irrigation. To ensuring that the effects of the different treatments are compared under the same environmental conditions, the goodness of adopting the data-driven decision-making approach is evaluate by comparing the final Panax notoginseng growth status of the respective fields.

Intergroup controls are also conducted between different fields: A1, B1, C1, D1, and E1. A data-driven decision-making-based program is implemented at different stages to evaluate the effectiveness of the intervention timing through the growth status of the post-intervention Panax notoginseng fields. This design allowed a more comprehensive assessment of the impact of the experimental treatments on the results and improved the scientific validity and reliability of the experiment.

## 3. Methods

### 3.1. Timing of Intervention

In order to investigate the accuracy of the intervention timing and the effectiveness of the intervention regulation on Panax notoginseng, the research steps in this paper are shown in Figure 7.

During the entire experiment, the choice of pre-intervention and post-intervention periods is an important feature of conditioning. In this scenario, it is essential to find a suitable timing for intervention. For example, Sun et al. [34] set up a multi-group orthogonal experiment. Intervention irrigation is regulated at different time periods, and the intervention time is obtained by comparing the state of the final crop. Thus, the intervention time is a hypothetical point in the time series after which the intervention regulator has been imposed. In order to predict the timing for intervention in advance, this paper proposes a type of combined framework called deep-learning-based intervention timing prediction, as shown in Figure 8.

First, the Informer model is employed to preprocess the time-series data and generate initial predictions. By leveraging the self-attention mechanism, the Informer model effectively captures long-range dependencies within the time series, thereby producing preliminary prediction results. Subsequently, the output of the Informer is fed into an LSTM (Long Short-Term Memory) network. The LSTM further refines and optimizes the predictions through its gating mechanisms, capturing more complex non-linear relationships in the time series. Finally, an Exponentially Weighted Moving Average (EWMA) is applied to smooth the LSTM-adjusted results, reducing noise and enhancing the stability of the predictions.

The Informer–LSTM–EWMA integrated model fully leverages the strengths of the Informer model and the LSTM model, combining long-term dependency modeling and multi-scale information fusion. By integrating the multi-scale information fusion capability of the Informer model with the long-term dependency modeling ability of the LSTM model, this integrated model can more comprehensively capture the features in time-series data. The EWMA model is capable of responding in real-time to incoming new data and continuously updating the smoothing coefficient to promptly reflect changes in the data as well as trends.

### 3.2. Model

#### 3.2.1. Informer

ProbSparse Self-Attention [35] is used to select important q−k pairs that describe the similarity between the *i*-th query and the key. The attention value of the *i* query to the key can be expressed as $p(k_{j}|q_{i})$ with probability. The closer the distribution *p* is to the uniform distribution *q*, the less important it is. The sparsity measurement *M* of the *i*-th query can be approximated as Equation (Equation 4).(4)$$\bar{M}\left(q_{i},K\right)=\max_{j}\left\{\frac{q_{i}k_{j}^{T}}{\sqrt{d}}\right\}-\frac{1}{L_{K}}\sum_{j=1}^{L_{K}}\frac{q_{i}k_{j}^{T}}{\sqrt{d}}$$

Based on the proposed metric, the number of $u=c\cdot \ln L_{Q}$ dominated queries is defined. ProbSparse self-attention can be defined as Equation (Equation 5).(5)$$A\left(Q,K,V\right)=Softmax\left(\frac{\bar{Q}K^{T}}{\sqrt{d}}\right)V$$
where $\bar{Q}$ is a sparse matrix that includes only the largest *u* queries in *M*, with *d* representing the input dimension.

Self-attention distilling [35] can significantly reduce the input dimension. Essentially, it is a tandem combination of *Conv1d*, *ELU* activation function, and maxpooling. The formula for advancing from layer *j* to layer (j+1) is shown in Equation (Equation 6).(6)$$X_{j+1}^{t}=Maxpooling\left(ELU\left(Conv1d\left({\left[X_{j}^{t}\right]}_{att}\right)\right)\right)$$
where ${\left[\cdot \right]}_{att}$ denotes an attention block.

#### 3.2.2. LSTM

LSTM is a deep-learning model for processing sequential data. It contains a storage unit and three gates: an oblivion gate, an input gate, and an output gate. These gates serve to selectively control the flow of information, allowing the model to capture long-term dependencies more efficiently in time sequences.

Each gate produces a state variable $i_{t}$, $f_{t}$, and $o_{t}$ at time *t* and an output unit $h_{t}$, respectively. The state update process is shown in Equations (Equation 7)–(Equation 12).(7)$$f_{t}=\sigma \left(w_{f}\left[h_{t-1},x_{t-1}\right]+b_{f}\right)$$(8)$$i_{t}=\sigma \left(w_{i}\left[h_{t-1},x_{t-1}\right]+b_{i}\right)$$(9)$${\tilde{C}}_{t}=\tanh\left(w_{c}\left[h_{t-1},x_{t-1}\right]+b_{c}\right)$$(10)$$o_{t}=\sigma \left(w_{o}\left[h_{t-1},x_{t-1}\right]+b_{o}\right)$$(11)$$C_{t}=C_{t-1}*f_{t}+i_{t}*{\tilde{C}}_{t}$$(12)$$h_{t}=o_{t}*\tanh\left(C_{t}\right)$$
where $\left[w_{f},b_{f}\right]$, $\left[w_{c},b_{c}\right]$, $\left[w_{i},b_{i}\right]$, and $\left[w_{o},b_{o}\right]$ correspond to the weight matrix and bias term of the forgetting gate $f_{t}$, cell state $C_{t}$, input gate $i_{t}$, and output gate $o_{t}$, respectively, at time *t*. $\left[h_{t-1},x_{t-1}\right]$ is the output in the hidden layer at time t−1 and the input at time *t*, respectively, and $\tilde{C}$ is the input node state.

Based on this mechanism, LSTM can retain important information in sequence data and perform prediction or classification tasks accordingly.

#### 3.2.3. EWMA

The EWMA model is a method used to estimate trends and cyclical variations in time-series data. It performs a weighted average of historical data to better capture the trend of recent observations. It is widely used for the smoothing and trend forecasting of time-series data. In an EWMA model, newer observations are given a larger weight while older observations are given a smaller weight. This allows the model to be more flexible in adapting to changes in the data and also reduces the impact of random noise on the predictions. The formula for the EWMA model is as in Equation (Equation 13).(13)$$S_{t}=\alpha Y_{t}+\left(1-\alpha \right)S_{t-1}$$
where $S_{t}$ is the EWMA value at time *t*. $Y_{t}$ is the observed value at time *t*. 0<α<1 is a smoothing parameter that determines the relative weight of the observation to the previous EWMA value.

The model parameters during training are set as follows: Input sequence length of the message encoder seq_len=164. Start token length of Informer decoder label_len=156. Prediction sequence length pred_len=150. Number of heads of attention mechanisms n_heads=8. The sampling factor c=5, batch_size=48. The number of training epochs epoch=9. This paper adopts the Adam optimizer.

### 3.3. Probability Density Function

Before the beginning of the experiment, the state of Panax notoginseng planted in each experimental field is quite different. The morphological data of some plants will be more prominent due to the individuality of the plants, but this does not affect the whole experiment. To provide an intuitive characterization of the spatial distribution patterns of Panax notoginseng crops, this study employs the Probability Density Function (PDF) for quantitative analysis. The methodological workflow comprises the following key steps:Step1: Histogram Construction and Normalization. The methodological workflow initiates by inputting the dataset to determine the bin count *p* for histogram configuration. Subsequently, normalization is performed to derive the Probability Density Function (PDF). The normalized value of each histogram bin is calculated according to Equation (Equation 14).(14)$${\mathrm{Bin}}_{i}=\frac{k_{i}}{K\times L}$$Let *K* denote the total number of data points, $k_{i}$ represent the number of data points within the i−th interval, *L* be the width of each interval, ${\mathrm{Bin}}_{i}$ indicate the value of the *i*-th histogram bin, and $\sum_{i=1}^{p}\left({\mathrm{Bin}}_{i}\times L\right)=1$Step2: Generation of Theoretical Distribution Coordinates. To provide plotting coordinates for subsequent theoretical distribution curves (e.g., the normal distribution) and ensure consistency with the histogram range, a linearly spaced abscissa vector is generated, as shown in Equation (Equation 15).(15)$$x=\left[x_{\min},x_{\min}+\Delta ,\ldots ,x_{\max}\right],\;\Delta =\frac{x_{\max}-x_{\min}}{g}$$In the formula, $x_{\max}$ denotes the maximum value of the dataset, $x_{\min}$ represents the minimum value of the dataset, and *g* specifies the number of equidistant points to be generated.Step3: Calculation of the Normal Distribution Probability Density Function. The Probability Density Function (PDF) of the normal distribution is given by Equation (Equation 16).(16)$$f\left(x;\mu ,\sigma \right)=\frac{1}{\sigma \sqrt{2\pi }}e^{-\frac{{\left(x-\mu \right)}^{2}}{2\sigma^{2}}}$$
where μ is the mean, which determines the center position of the distribution, and σ is the standard deviation, which controls the width of the distribution (the larger the σ, the flatter the curve).

Due to too few bins (<10) masking the details and too many bins (>30) leading to significant noise, the number of bins in this paper is set to f=20, with the number of equidistant points g=100.

### 3.4. Prediction

Before the experiment, it is essential to ensure that the crop conditions in each experimental plot are consistent to guarantee the fairness and comparability of the experiment. This is achieved through the following measures:*Crop condition consistency:* In the early stages of the experiment, the growth status of crops in all experimental plots is measure and assessed to ensure they are at similar levels;*Environmental condition control:* The soil type, fertility levels, and other environmental factors of each experimental plot are kept as consistent as possible to eliminate external factors that could interfere with the results;*Randomized design:* A randomized grouping method is employ to assign the experimental plots to different treatment groups, further reducing potential biases.

Through these measures, this experiment ensures consistency in initial conditions, laying a foundation for the reliability and scientific validity of the subsequent results.

In the Panax notoginseng growth intervention experiment, we predict the growth condition of Panax notoginseng in the next cycle. The aim is to intervene early before the growth transitions down to ensure normal growth of Panax notoginseng.

When prediction processes run, the historical data collected from the above features are used as inputs to the model, and the crop height pattern data are used as outputs. During the experiment, we only need to know the growth status of the crop; we do not need to know the exact value of the morphological data such as plant height. When a prediction is carried out, we normalize the output data so that its output is in the range of [0,1].

Similarly, in the Panax notoginseng experimental field, we use previously collected historical datasets to train a model to predict the growth performance of Panax notoginseng in the coming week. The model can predict the growth status of the crop for the coming growth period. We have the following expectations for the outcome of the forecast:When we make a growth prediction for Panax notoginseng, our main purpose is not to predict the real status but to obtain the time of intervention.The model should have the ability to generalize and be able to make predictions, even in different experimental sites.The prediction of the intervention timing should not be later than the point at which the actual decline in the growth state of Panax notoginseng occurs. Due to the characteristics of the crop, delayed intervention may lead to irreversible damage or even death of the plant.

In order to illustrate the rationality of our method, we choose the LSTM, CNN-LSTM, and BiLSTM models as a comparison. The prediction of intervention timing is shown in Figure 9.

As evidenced by the comparative analysis in Figure 9, the Informer–LSTM–EWMA model provides approximately 5-day advance warnings for irrigation scheduling prior to crop status deterioration, while other forecasting models show different lag (after multiple predictions, our model is able to give the irrigation time 3–5 days before the crop condition declined). According to the predicted irrigation time, the manager can intervene by timely water supply when the plant does not appear to have irreversible water stress, so as to maintain the normal physiological metabolism of Panax notoginseng.

## 4. Results

### 4.1. Forecasting and Irrigation Records

When we perform prediction and irrigation, it is completely different from the model training. During the experiment, the real growth situation of Panax notoginseng in the future is completely unknown. In order to guide irrigation more accurately, a dataset is collect from about 5 days after the end of the last irrigation. We build the prediction model based on the dataset and make three repeated predictions, and then take the average of the three as the final result.

Taking the irrigation period from 1 May to 9 May 2024 as an example, after the irrigation is complete on 1 May, we need to wait five days and collect the data during this period. On 6 May, the collected historical data are input into the model to obtain the next irrigation time, as shown in Figure 10.

As can be seen from Figure 10, the three predicted results based on the same data collected from 1 May to 5 May are 71, 69, and 73, respectively. By averaging, it can be concluded that the next irrigation time will be about 71 h later, that is, three days later. The next irrigation will take place on 9 May. The same is true for subsequent irrigation operations, and irrigation time records are shown in Table 6.

In order to verify the differences in crop-growth state among the proposed method, the traditional method, and the combined methods compared with the two methods mentioned above, we adopt different programs for each experimental field, and the irrigation records of the experimental fields during the experimental period are shown in Table 6 and Table 7. Table 6 shows the irrigation records by the proposed method in this paper, and Table 7 shows the irrigation records by the traditional method. Moreover, the data-driven irrigation is applied in experimental sites B1, C1, and D1 at the first irrigation time, and then it is replace by the traditional method at different sampling times.

Taking experimental fields B1 and C1 as an example, experimental fields B1 and C1 are irrigated by data-driven irrigation during the irrigation period from 23 April to 9 May. In the irrigation period after 9 May, the irrigation mode of experimental site C1 changed to the traditional irrigation mode (as shown in the irrigation record of 13 May in Table 7), while the data-driven irrigation mode is normally adopted in experimental site B1 (as shown in the irrigation record of 19 May in Table 6). Therefore, there is no irrigation record of experimental field C1 after 9 May in Table 6, but there is a corresponding irrigation record on 13 May in Table 7. As can be seen from Figure 6, the area of experimental field C1 is about half that of experimental field C2, so the irrigation of C1 after replacement by the traditional irrigation method is also about half that of C2, that is, about 70L.

During the experiment, when the model gives the corresponding prediction irrigation time, we will irrigate experimental sites according to the predicted results. Considering that it takes time for water to permeate, we chose data from the second day after irrigation as the state of the soil environment. The humidity data before and after the data-driven irrigation method is shown in Table 8 and Table 9, where the average value of the data collected by the corresponding sensor node for three consecutive time instance is computed as the final result.

In Table 9, “/” indicates that humidity data are not recorded because data-driven irrigation is replaced by the traditional one on this time instance.

### 4.2. Effects of Varied Irrigation Approaches on Spatial Distribution Patterns of Panax Notoginseng in Field Conditions

The distribution and probability density function of the height of Panax notoginseng plants in each experimental field before starting the experiment are shown in Figure 11 (recorded on 8 April 2024). The distribution and probability density function of Panax notoginseng plant height in the experimental group at the end of the experiment are shown in Figure 12. Actual pre- and post-experiment comparisons of partial field sites are shown in Figure 13.

Comparative analysis of subfigures a and b in Figure 11 and Figure 12 reveals that crops subjected to data-driven irrigation methods exhibit significantly concentrated plant height distributions. Further observation of subfigures c, d, and e in Figure 11 and Figure 12 demonstrates that crops irrigated predominantly by traditional methods display distinct discrete distribution patterns in plant height. This contrast visually demonstrates the superior homogenization control capability of data-driven irrigation strategies over conventional approaches in crop-growth regulation.

As shown in Figure 13, panels a, b, c, and d represent the actual situation of crops before the experiment. Panels e and f represent the situation of crops after the experiment with data-driven irrigation. Panels g and h represent the situation of crops after the experiment with traditional irrigation. Through the comparison before and after the experiment, the crop density using the traditional irrigation method showed a serious decline. The reason for this phenomenon is that irrational irrigation leads to the death of Panax notoginsen. The temporal variations in mean plant height and population dynamics of experimental crops are systematically presented in Section 4.3 and Section 4.4 of this study.

### 4.3. Effects of Varied Irrigation Approaches on Mean Plant Height in Panax Notoginseng

During the nearly two-month experiment, the average plant height growth of the crops in the experimental filed is shown in Figure 14. We repeated the measurements three times and took the average as the recorded height of each plant. Then, the average plant height of the whole experimental plot is calculated.

The data-driven irrigation strategy is adopted in the whole growth process of A1, while the traditional fixed irrigation volume strategy is adopted in the whole growth process of E1. As can be seen from Figure 14, the average plant height of E1 crops showed a growing trend in the early stage, but the growth state of crops kept declining in the later stage. A1 and B1 are in a good state of growth, B1 underwent a period of intervention regulation in the late stage, although in a growing trend, but its growth is not as good as A1. The growth state of D1 and C1 experiment fields has been in a good state after the data-driven intervention irrigation strategy in the early stage, but the growth state showed a significant decline after stopping intervention from the fourth time and the third time, respectively.

The growth of the average plant height of Panax notoginsen in the control group between the groups is shown in Figure 15.

The five experimental fields (A2, B2, C2, D2, and E2) are irrigated with traditional irrigation methods, as shown in Table 7. The experimental results showed that the data-driven irrigation strategy is still superior to the traditional irrigation method under the same soil conditions. The influence of soil conditions is excluded by means of a control between groups.

According to the data comparison in Figure 14 and Figure 15, the average plant height in field A1 using data-driven irrigation increased significantly from 192 mm before the experiment to 212 mm. In contrast, the average plant height in field E1 using traditional irrigation decreased from 188 mm to 175 mm. This difference is mainly attribute to the fact that, under traditional irrigation, some initially well-growing crops died due to improper watering, thereby affecting overall crop quality. To rule out the influence of soil factors, the study establish field A2 as an internal control for A1. The results showed that the average plant height decreased from 191 mm to 174 mm in field A2, where traditional irrigation is applied. Traditional irrigation strategies are adopted in experimental fields B2, C2, D2, and E2. As can be seen from Figure 15, the average crops in experimental fields B2, C2, D2, and E2 showed a downward trend, further confirming the impact of irrigation methods on crop growth.

According to the irrigation records in Figure 14 and Table 6 and Table 7, it can be found that experiments B1, C1, and D1 replace the data-driven irrigation method by the traditional one at different irrigation times. Then, crop-growth conditions all decline to varying degrees. After the irrigation method is replaced in experiments C1 and D1, the average plant height in experiment field C1 dropped from 212 mm on 13 May to 194 mm on 21 May, and the average plant height in experiment field D1 dropped from 213 mm on 6 May to 206 mm on 13 May. Taking experimental field C1 as an example, it can be seen from Table 6 and Table 7 that experimental field C1 switched to the traditional irrigation mode on 13 May after the data-driven irrigation mode on 9 May, at which time the humidity data of sensor node 1 increased to 42.1%. Because of the switch of irrigation methods, the soil moisture reached a range that is not conducive to the growth of Panax notoginseng, resulting in the death of some crops that had been growing well during this period. Although the data-driven irrigation strategy is replaced in field B1 after 20 May, the average plant height of crops did not decrease or even increased by 1 mm between 27 May and 3 June, but the increase is less than the increase of 2 mm in field A1 where data-driven irrigation is adopted at this stage. These results show that our irrigation method including prediction time and data-driven irrigation mechanisms are effective.

Compared with the traditional irrigation method, the average plant height of Panax notoginseng based on the data-driven intervention irrigation method increased by about 21.8%, and the cumulative irrigation times decreased once.

### 4.4. Effects of Varied Irrigation Approaches on Plant Population in Panax Notoginseng

Based on the analysis of the final survival count of the crops, the experimental data, as presented in Table 10, indicate that Experimental Group 1 initially consisted of 33 plants, which increased to 87 by the conclusion of the experiment. Within this group, the internal control subgroup A1 grew from 10 to 51 plants, while the experimental subgroup A2 increased from 23 to 36 plants, yielding a growth rate of 56.5%. In contrast, Control Group 5, which started with 36 plants, reached a total of 54 plants by the end of the experiment, corresponding to a growth rate of 50.0%.

## 5. Discussion

The proposed model effectively integrates deep learning with the Exponentially Weighted Moving Average (EWMA) method by utilizing historical data and real-time monitoring data. This integration enables accurate prediction of crop status decline trends 3–5 days in advance, allowing for timely irrigation warning signals to be sent to administrators. In contrast, traditional prediction methods, such as the LSTM-EWMA, CNN-LSTM-EWMA, and BiLSTM-EWMA models, exhibit significant latency, often issuing warning signals only after crops have already shown signs of status deterioration. This delayed warning may result in missing the optimal irrigation window, prolonging the period of water stress in crops and thereby affecting their normal physiological and metabolic processes [36].

Specifically, water stress can inhibit the photosynthetic efficiency of crops, reduce the rate of dry matter accumulation, and hinder plant growth [37]. More critically, prolonged water stress may lead to irreversible damage to crops. Therefore, the timely warning function of the Informer–LSTM–EWMA model is of practical significance for ensuring normal crop growth and maintaining stable yields, providing reliable technical support for the implementation of precision agriculture.

Extensive research indicates that rational irrigation strategies positively influence the alteration of crops’ physiological and ecological traits [38,39,40]. Contemporary studies suggest that both low-level and high-level irrigation could detrimentally impact plant development [41]. This research validates earlier findings, and data from control experimental field 5 substantiate that inappropriate irrigation methods negatively influence crop productivity.

The experimental findings substantiate the superiority of the proposed irrigation strategy over conventional methods, which is manifested in three key dimensions:(a)*Spatial Distribution Pattern:* Figure 11, Figure 12 and Figure 13 illustrate that crops under the proposed irrigation system exhibit a more clustered distribution compared to the dispersed and sparse arrangement observed in traditional irrigation practices.(b)*Mean Plant Height:* The data presented in Figure 14 and Figure 15 demonstrate a significant 21.8% increase in average plant height when using the proposed irrigation method relative to the control group.(c)*Crop Density and Growth Rate:* As evidenced by Table 10, the proposed irrigation approach yields higher final crop density and enhanced growth rates compared to traditional irrigation techniques.

Precision irrigation offers substantial socio-economic and environmental benefits, reducing water and energy use while boosting crop yields [42]. Cotton irrigation trials confirm its effectiveness in enhancing production [43]. Developing a predictive platform can provide optimal irrigation strategies, further improving agricultural outcomes [44].

As can be seen before and after the experiment, the use of data-driven irrigation can effectively guide farmers to take effective interventions in the field. The method ensures that the crop is in a healthy state, and the method has the following benefits:The method does not predict the true value, but predicts the time of intervention when the crop will need to be intervened. This avoids growth uncertainty caused by different crops for their own reasons.Data-driven irrigation is more scientific than traditional experiential irrigation, and in the context of modern smart agriculture, a data-driven approach makes it easier to create a plant factory that increases yields.The method is not only applicable to Panax notoginseng but can also be used in the cultivation of other economic crops. This method greatly helps managers obtain higher quality and yield with less energy input.

## 6. Conclusions

In this study, our proposed data-driven irrigation-based approach can notify managers and guide them in time according to the growth trend of Panax notoginseng. During the experiment, we conducted a comprehensive evaluation by the average plant height of the crop in the experimental site and the distribution of the crop height. The final results show that the proposed method is much better than the traditional fixed irrigation method in the greenhouse environment. It is found that the optimal amount of irrigation required by the crop is different at different times. For example, the current period of irrigation may be sufficient for the crop to maintain optimal growth, but the same amount of irrigation in the next period may inhibit the growth of the crop to the point of death. In conclusion, we believe that this method is applicable to agriculture because it adds an interpretable component to existing plant-growth models by predicting crop-growth status in real time and is sensitive to changes in crop-growth trends due to different irrigation treatment conditions, and the method could theoretically be extended to other crop cultivation. In order to obtain a comprehensive crop-growth trend prediction model on a larger scale, it is necessary to increase the amount of data (such as weather, precipitation, wind speed, etc.). In modern agriculture, smart irrigation is very important for the development of agriculture. Various modern irrigation strategies have been successfully applied to the cultivation of crops, and in the future data-driven-based irrigation will be an effective method to study smart agriculture and can improve crop yields.

## Figures and Tables

**Figure 1 plants-14-01226-f001:**
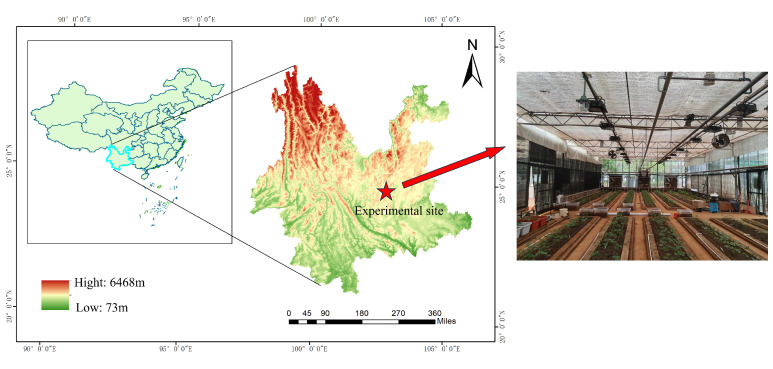
Experimental environment.

**Figure 2 plants-14-01226-f002:**
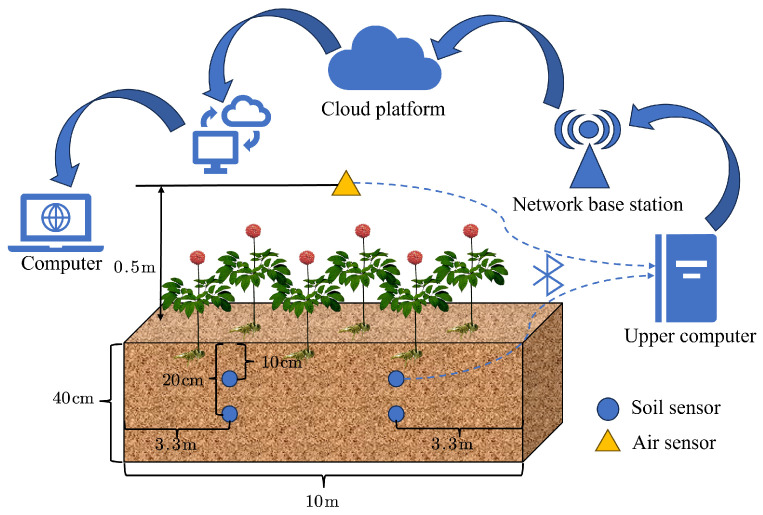
Data acquisition procedure.

**Figure 3 plants-14-01226-f003:**
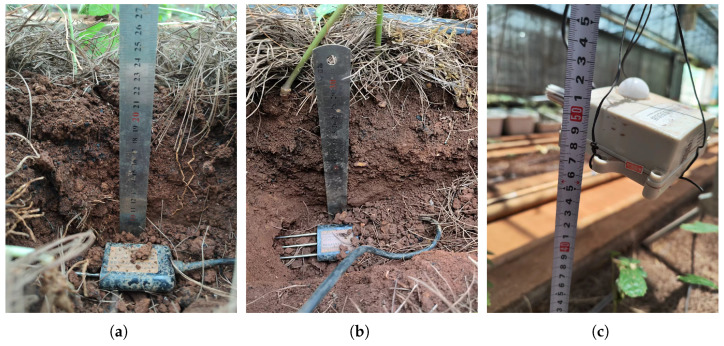
Sensor placement methods in experimental fields. (**a**) Sensor installation at 10 cm soil depth. (**b**) Sensor installation at 20 cm soil depth. (**c**) Air sensor installation.

**Figure 4 plants-14-01226-f004:**
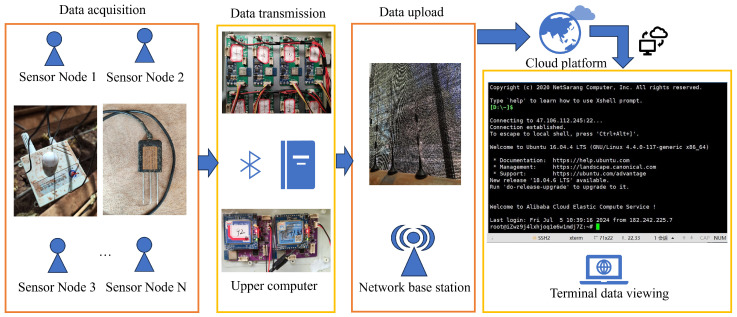
Data collection workflow.

**Figure 5 plants-14-01226-f005:**
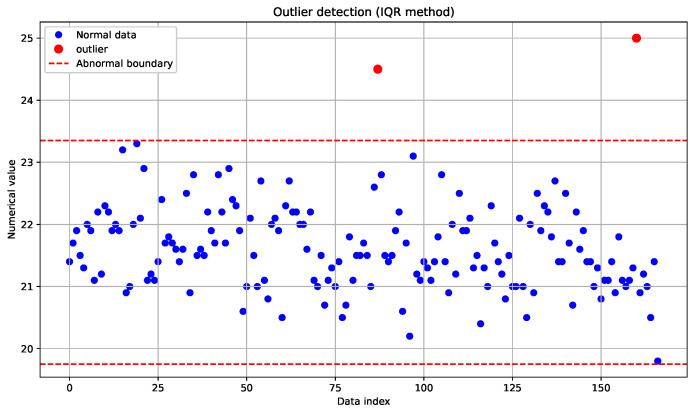
Outlier detection.

**Figure 6 plants-14-01226-f006:**
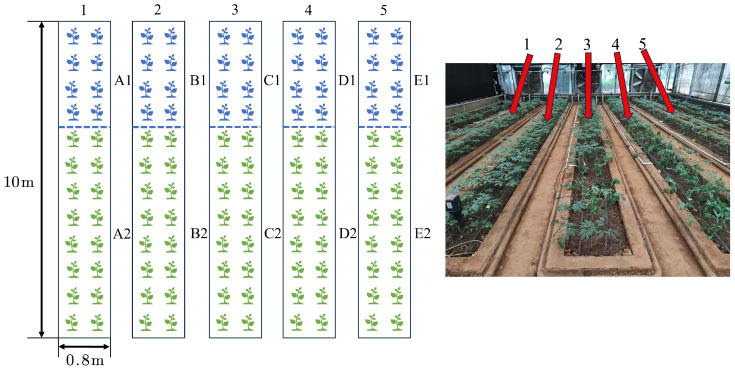
Schematic diagram of the experimental field demarcation.

**Figure 7 plants-14-01226-f007:**
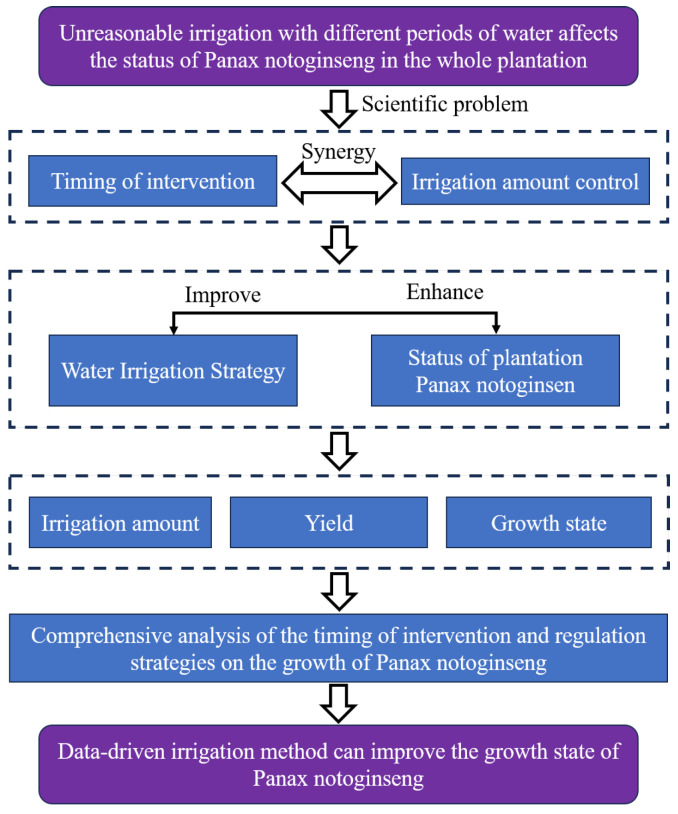
Research flowchart.

**Figure 8 plants-14-01226-f008:**
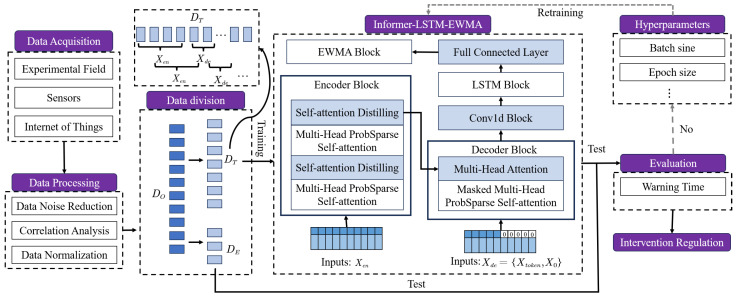
Methodology flow chart. $D_{O}$, $D_{T}$, and $D_{E}$ denote the original dataset, original training set, and test set, respectively. $X_{en}$ is the input to the encoder module, which is historical time-series data and external features. $X_{de}$ is the input to the decoder module and is a vector consisting of a historical time-series data vector $X_{token}$ and a vector consisting of a target time-series data vector $X_{0}$.

**Figure 9 plants-14-01226-f009:**
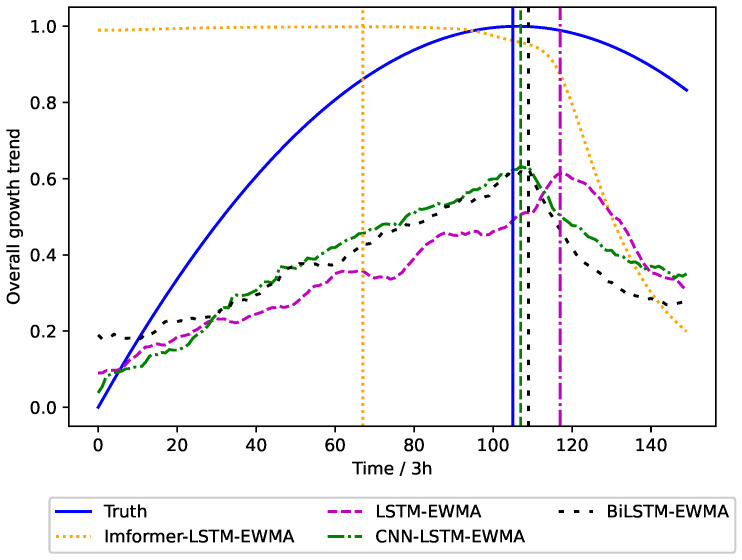
Different models predict time to intervention.

**Figure 10 plants-14-01226-f010:**
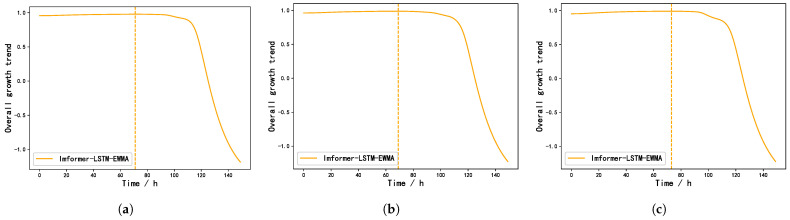
Forecast results for the period from 1 May to 9 May 2024. (**a**) First forecast results. (**b**) Second forecast results. (**c**) Third forecast results.

**Figure 11 plants-14-01226-f011:**
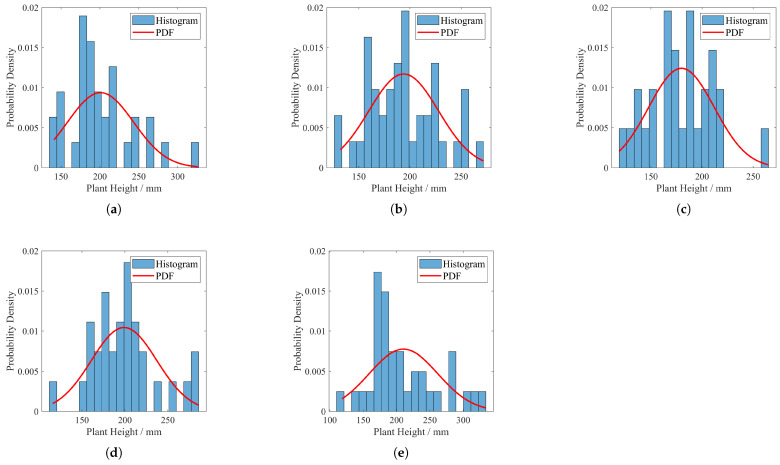
Plant height distribution in each experimental field before the experiment. (**a**) Experimental 1. (**b**) Experimental 2. (**c**) Experimental 3. (**d**) Experimental 4. (**e**) Experimental 5.

**Figure 12 plants-14-01226-f012:**
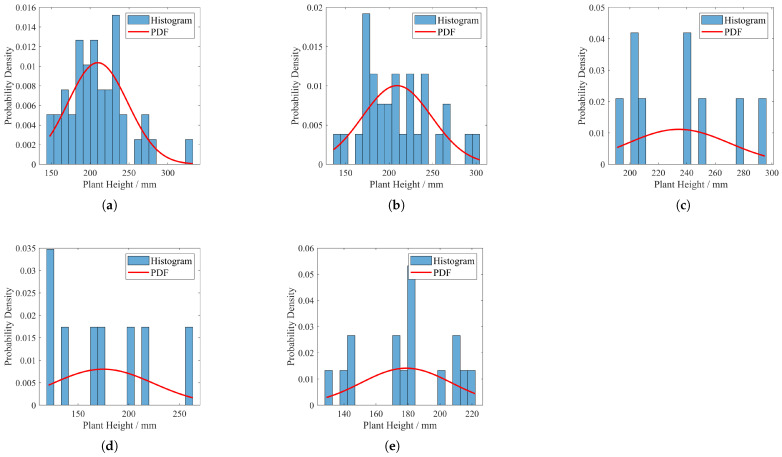
Plant height distribution in each experimental field after the experiment. (**a**) Experimental 1. (**b**) Experimental 2. (**c**) Experimental 3. (**d**) Experimental 4. (**e**) Experimental 5.

**Figure 13 plants-14-01226-f013:**
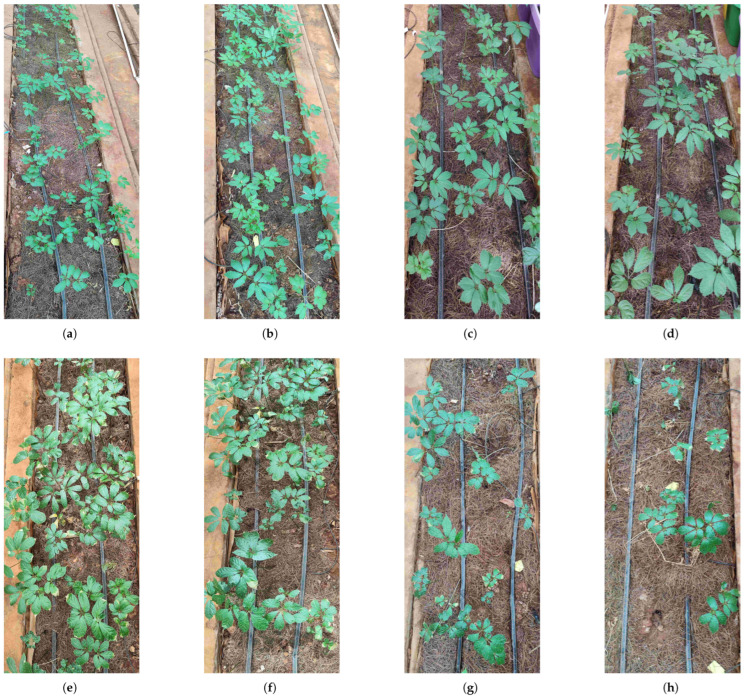
Growth of Panax notoginseng using data-driven irrigation strategy and traditional irrigation strategy. ((**a**–**d**) represent the actual condition of Panax notoginseng in the experimental field before the experiment; (**e**,**f**) represent the growth condition after adopting the data-driven irrigation strategy in the experimental field (**a**,**b**); and (**g**,**h**) represent the growth condition after adopting traditional irrigation strategy in experimental field (**c**,**d**)).

**Figure 14 plants-14-01226-f014:**
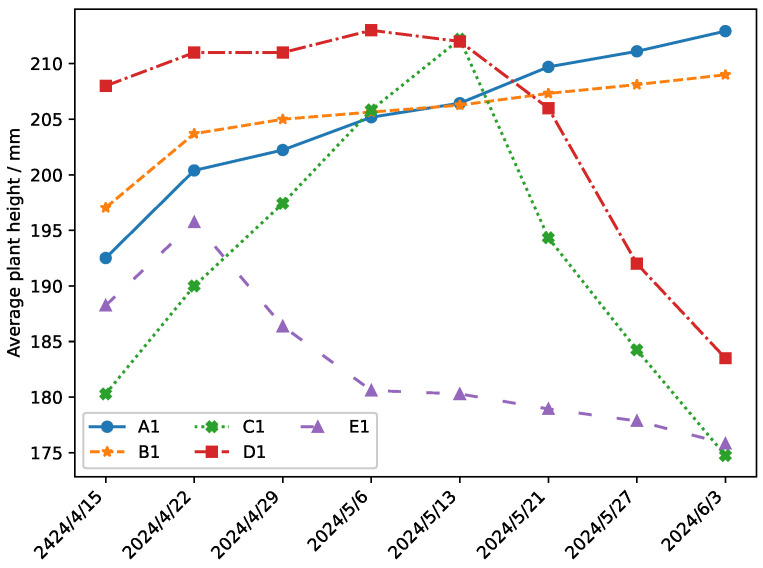
The growth of the average plant height in the experimental field.

**Figure 15 plants-14-01226-f015:**
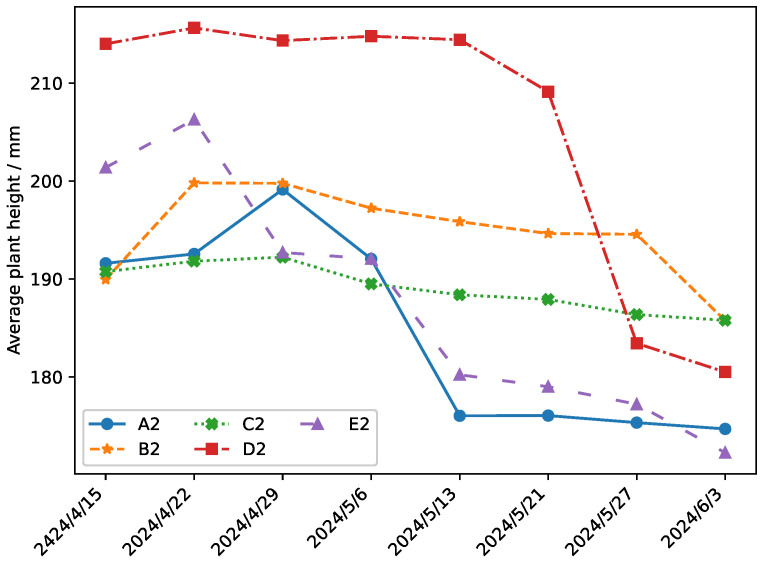
The growth of the average plant height of the control group.

**Table 1 plants-14-01226-t001:** The transpiration rate of the crops.

Experimental Field	Transpiration Rate (mmol·m^−2^·s^−1^)
Field 1	1.31±0.15f
Field 2	1.49±0.26d
Field 3	1.93±0.13a
Field 4	1.24±0.16e
Field 5	1.35±0.17c

Data are present as the mean of three replicates. Values are shown as “mean ± standard deviation”, and different lowercase letters denote significant differences among treatments at the 0.05 level.

**Table 2 plants-14-01226-t002:** Sensor parameter table.

Parameters	Range	Resolution	Precision
Soil conductivity	0~20,000 us/cm	10 us/cm	±3%
Soil moisture	0~100	0.1%	2%
Soil temperature	−40~80 °C	0.1 °C	±0.5 °C
Air temperature	−40~80 °C	0.1 °C	±0.5 °C
Air humidity	0%*RH*~80%*RH*	0.1%	±3%RH

**Table 3 plants-14-01226-t003:** Technical characteristics of the air sensor.

Indicators	Description
DC power supply (default)	10–30 VDC
Maximum power consumption	0.4 W
Output signal	RS485 (Modbus protocol)
Manufacturer	Shandong Renke measurement and control Technology Co., Ltd. (Jinan, China)

**Table 4 plants-14-01226-t004:** Technical characteristics of the soil sensor.

Indicators	Description
DC power supply (default)	DC 4.5–30 V
Maximum power consumption	0.7 W
Output signal	RS485 (Modbus protocol)
Class of protection	IP68
Manufacturer	Shandong Renke measurement and control Technology Co., Ltd. (Jinan, China)

**Table 5 plants-14-01226-t005:** Treatments in different experimental fields.

Experimental Field	Stage 1	Stage 2	Stage 3	Stage 4
A1	✓	✓	✓	✓
B1	✓	✓	✓	✗
C1	✓	✓	✗	✗
D1	✓	✗	✗	✗
E1	✗	✗	✗	✗

✓ indicates the use of a data-driven approach for intervention regulation at this stage. ✗ indicates that no modality of intervention regulation will be used at that stage.

**Table 6 plants-14-01226-t006:** Irrigation records from different experimental fields.

Time	Field A1	Field B1	Field C1	Field D1	Field E1
04-23	70L	85L	90L	80L	∖
05-01	80L	90L	85L	85L	∖
05-09	80L	95L	70L	∖	∖
05-19	85L	95L	∖	∖	∖
05-27	90L	∖	∖	∖	∖

In the table, ∖ indicates that the traditional irrigation method in Table 7 is used instead of intervention irrigation.

**Table 7 plants-14-01226-t007:** Irrigation records of the control group.

Time	Field A2	Field B2	Field C2	Field D2	Field E2	Field B1	Field C1	Field D1	Field E1
04-22	135L	135L	135L	135L	135L	∖	∖	∖	70L
04-29	135L	135L	135L	135L	135L	∖	∖	∖	70L
05-06	135L	135L	135L	135L	135L	∖	∖	70L	70L
05-13	135L	135L	135L	135L	135L	∖	70L	70L	70L
05-20	135L	135L	135L	135L	135L	∖	70L	70L	70L
05-27	135L	135L	135L	135L	135L	70L	70L	70L	70L

In the table, ∖ indicates irrigation in the manner shown in Table 6.

**Table 8 plants-14-01226-t008:** Humidity data of sensor before irrigation.

Time	Field A1/%RH		Field B1/%RH		Field C1/%RH		Field D1/%RH		Field E1/%RH	
	Node 1	Node 2	Node 1	Node 2	Node 1	Node 2	Node 1	Node 2	Node 1	Node 2
04-23	21.5	22.0	22.5	26.0	20.3	18.9	25.1	27.1	25.7	26.5
05-01	23.9	25.9	20.1	23.6	22.4	25.1	24.2	28.8	26.2	30.6
05-09	26.1	29.6	19.1	20.9	26.6	29.2	25.3	28.4	29.5	31.2
05-19	26.8	28.3	26.5	29.6	25.5	28.6	26.1	29.3	28.4	30.9
05-27	20.6	24.3	28.2	30.4	26.2	29.4	27.6	31.2	30.1	33.2

Node 1 is the sensor at 10 cm from the soil plane and Node 2 is the sensor at 20 cm from the soil plane.

**Table 9 plants-14-01226-t009:** Humidity data of sensor after irrigation.

Time	Field A1/%RH		Field B1/%RH		Field C1/%RH		Field D1/%RH		Field E1/%RH	
	Node 1	Node 2	Node 1	Node 2	Node 1	Node 2	Node 1	Node 2	Node 1	Node 2
04-24	26.7	28.9	29.0	30.8	30.8	32.1	29.6	33.2	/	/
05-02	30.4	35.8	33.1	36.4	34.8	36.7	30.7	34.1	/	/
05-10	33.9	36.2	35.9	38.6	29.2	33.9	/	/	/	/
05-20	34.5	38.3	38.4	40.1	/	/	/	/	/	/
05-28	34.6	39.1	/	/	/	/	/	/	/	/

Node 1 is the sensor at 10 cm from the soil plane and Node 2 is the sensor at 20 cm from the soil plane.

**Table 10 plants-14-01226-t010:** Crop survival count record sheet (number of plants).

Time	Field A1	Field A2	Field E1	Field E2
04-22	10	23	12	24
04-29	27	21	12	22
05-06	31	29	14	27
05-13	33	30	15	31
05-20	42	33	16	34
05-27	51	36	18	36

## Data Availability

Data can be obtained from authors via email.

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
