# Peer review of "Growth Trend Prediction and Intervention of Panax Notoginseng Growth Status Based on a Data-Driven Approach"

_plants, 2025, doi:10.3390/plants14081226_

Round 1

Reviewer 1 Report

Comments and Suggestions for Authors

This study employs an Informer-LSTM-EWMA model to predict crop growth states, enabling data-driven irrigation decisions based on future forecasts. The paper can be published after the following minor revision:

1-The literature review presented in the article should be expanded especially on irrigation strategies.
2- The research gap should be clearly stated. What makes this study better than previous approaches?
3- The novelty of the proposed methodology should be emphasized.
4- More discussions should be conducted to show advantages of the proposed method.
5- The authors should justify why they chose Informer-LSTM-EWMA over alternative models. How the LSTM-EWMA model works and why this model was chosen should be explained in more detail.
6- The data preprocessing section should be detailed. Information should be provided about filtering techniques and how missing values are handled.
7- Hyperparameter tuning strategies should be discussed and information about model training details should be provided (e.g., number of epochs, optimizer used).
8- The quality of figures (Figure 9,10 and 11) of the paper needs to be improved.

Reviewer 2 Report

Comments and Suggestions for Authors

The manuscript is not acceptable due to a series of poorly structured segments in the methodology and research description. Given the very poorly described and incomplete methodology, I suggest rejecting the manuscript for publication.

In short, the abstract is not well-structured and contains many unnecessary details (e.g., the number of plants per treatment). The end of the introduction, where the structure of individual sections is described, is also unacceptable.

In the Materials and Methods section, the climate description is inadequate and should use a recognized climate classification. Additionally, lines 105–110 are contradictory; the authors state that the research was conducted in an open field but later conclude it was in a greenhouse. The description of the installed sensors is incomplete, lacking details on the type of sensors, manufacturer, etc. It is unclear what the research treatments were and how they were controlled. Furthermore, the statistical analysis of the data is not presented at all.

Comments on the Quality of English Language

must be improved

Reviewer 3 Report

Comments and Suggestions for Authors

The article entitled ‘’Growth trend prediction and intervention of Panax notoginseng growth status based on data-driven approach” is in line with the scope of the journal. The content analysis reveals that the research offers relevant contributions to the area. However, there are some aspects that could be improved. For example, it would be important to include a discussion section so that the results and statements presented are properly substantiated. Below, I present more suggestions for improving the work.
In the title, add …status based on a data-driven approach
1. …irrigation has to be adjusted according…
7.. training an Informer-LSTM-EWMA model... Give the meaning of LSTM-EWMA?
11. …3−5 days in advance before.
23... Change the keywords. They should be different from the words in the article title. Suggestion: Irrigated cultivation, smart agriculture,
26 ..In order to To promote modern…
28...modernization policies is important for..
95-99. Remove lines 95 to 99 from the text. The sentence is not necessary. The article is divided into sections.
100. Materials and Programmatic. Remove the word Programmatic from the section title.
102. Provide the geographic coordinates (latitude, longitude and altitude) of the study area. It is interesting to use a map located in the study area in China.
112. In order to collect environmental data of air and soil... Provide the acquisition time. Are these daily average values? Please explain Figure 2 better. Provide the technical characteristics of the sensors (manufacturer).
123-125. Mention the noise associated with the data. Explain the variables of Formula 1.
129. All plants within the experimental plots were systematically measured. Specify better: the distance between rows of plants; number of plants per row. The age of the plants.
135. ...a fixed amount of water irrigation every week... Specify the amount of water irrigated daily or weekly. What irrigation system is used? Is it drip irrigation? Please inform the average evapotranspiration of the crop. It would be very important to have the typical value of the crop's evapotranspiration for reference in future work, even if you did not use the water demand in the experiment!
125, 197, 199, 204, 212 and 231. Change the word formula to Equation.
235. Experiments and results. Leave only the word Results. In this section, you mixed the description of the experiment with results. This description should explain Figure 2. Isn't it? I suggest separating experiment from results. You are describing the experiment from lines 236 to 252.
252. The distribution and probability density function (PDF) of the height of Panax notoginseng plants in each experimental plot before starting the experiment are shown in Figure 8. This is a result!
256-266. Description of the experiment! This is a method!
269-284. Prediction refers to a description of the method! It is not a result.
285-289. Would Figure 9 be a result? I think so!
289-302.... This model, leveraging history... You discuss the models. Please take this to the Discussion section and cite related articles to support your claims!
304. Create a new title or remove Result and Discussion. It is important to separate Results from Discussions. So, create a dedicated section for Discussion and relate to other articles to support your claims! 312. Insert the units of measurement of the variables in tables 3, 4, 5 and 6.
355. Figure 13 is the same as Figure 8. Please review this and improve the explanation of the results.
356-368. The sentences refer to the discussion section!
373-375. The final results show that the proposed method is much better than the traditional fixed irrigation method in the greenhouse environment. I did not see this! Please show this in the discussion section of the results so that it supports this statement!

Comments on the Quality of English Language

The article entitled ‘’Growth trend prediction and intervention of Panax notoginseng growth status based on data-driven approach” is in line with the scope of the journal. The content analysis reveals that the research offers relevant contributions to the area. However, there are some aspects that could be improved. For example, it would be important to include a discussion section so that the results and statements presented are properly substantiated. Below, I present more suggestions for improving the work.
In the title, add …status based on a data-driven approach
1. …irrigation has to be adjusted according…
7.. training an Informer-LSTM-EWMA model... Give the meaning of LSTM-EWMA?
11. …3−5 days in advance before.
23... Change the keywords. They should be different from the words in the article title. Suggestion: Irrigated cultivation, smart agriculture,
26 ..In order to To promote modern…
28...modernization policies is important for..
95-99. Remove lines 95 to 99 from the text. The sentence is not necessary. The article is divided into sections.
100. Materials and Programmatic. Remove the word Programmatic from the section title.
102. Provide the geographic coordinates (latitude, longitude and altitude) of the study area. It is interesting to use a map located in the study area in China.
112. In order to collect environmental data of air and soil... Provide the acquisition time. Are these daily average values? Please explain Figure 2 better. Provide the technical characteristics of the sensors (manufacturer).
123-125. Mention the noise associated with the data. Explain the variables of Formula 1.
129. All plants within the experimental plots were systematically measured. Specify better: the distance between rows of plants; number of plants per row. The age of the plants.
135. ...a fixed amount of water irrigation every week... Specify the amount of water irrigated daily or weekly. What irrigation system is used? Is it drip irrigation? Please inform the average evapotranspiration of the crop. It would be very important to have the typical value of the crop's evapotranspiration for reference in future work, even if you did not use the water demand in the experiment!
125, 197, 199, 204, 212 and 231. Change the word formula to Equation.
235. Experiments and results. Leave only the word Results. In this section, you mixed the description of the experiment with results. This description should explain Figure 2. Isn't it? I suggest separating experiment from results. You are describing the experiment from lines 236 to 252.
252. The distribution and probability density function (PDF) of the height of Panax notoginseng plants in each experimental plot before starting the experiment are shown in Figure 8. This is a result!
256-266. Description of the experiment! This is a method!
269-284. Prediction refers to a description of the method! It is not a result.
285-289. Would Figure 9 be a result? I think so!
289-302.... This model, leveraging history... You discuss the models. Please take this to the Discussion section and cite related articles to support your claims!
304. Create a new title or remove Result and Discussion. It is important to separate Results from Discussions. So, create a dedicated section for Discussion and relate to other articles to support your claims! 312. Insert the units of measurement of the variables in tables 3, 4, 5 and 6.
355. Figure 13 is the same as Figure 8. Please review this and improve the explanation of the results.
356-368. The sentences refer to the discussion section!
373-375. The final results show that the proposed method is much better than the traditional fixed irrigation method in the greenhouse environment. I did not see this! Please show this in the discussion section of the results so that it supports this statement!

Reviewer 4 Report

Comments and Suggestions for Authors

Dear Editor,
The manuscript requires adjustments and the suggestions, corrections and considerations are included in the attached file.

Comments on the Quality of English Language

The English is great.

Round 2

Reviewer 2 Report

Comments and Suggestions for Authors

The manuscript still remains unacceptable as the authors have not addressed the major concerns raised in the previous review. The methodology and research description remain poorly structured, incomplete, and lack crucial details necessary for scientific rigor. Given these fundamental shortcomings, I recommend rejecting the manuscript for publication.

Most critically, the statistical analysis is still entirely missing, despite previous comments highlighting this as a major flaw. Without a clear and transparent statistical approach, the reliability of the results cannot be assessed. The lack of any attempt to rectify this issue suggests that the manuscript does not meet the basic standards for publication.

Given the persistent deficiencies, I strongly advise against accepting this manuscript.

Comments on the Quality of English Language

The manuscript still remains unacceptable as the authors have not addressed the major concerns raised in the previous review. The methodology and research description remain poorly structured, incomplete, and lack crucial details necessary for scientific rigor. Given these fundamental shortcomings, I recommend rejecting the manuscript for publication.

Most critically, the statistical analysis is still entirely missing, despite previous comments highlighting this as a major flaw. Without a clear and transparent statistical approach, the reliability of the results cannot be assessed. The lack of any attempt to rectify this issue suggests that the manuscript does not meet the basic standards for publication.

Given the persistent deficiencies, I strongly advise against accepting this manuscript in its current form.

Author Response

Details can be seen in attechments.
